Impact of parental marital status on self-harm in Chinese primary school students: the mediating role of depression and the moderating effect of classmate relationships

Ai Ming 1 2
Xu Xiao-Ming 1 2
Wang Wo 3
Chen JianMei 1 2
He Jinglan 1 2
Zhang Qi 1 2
Hong Su 1 2
Gan Yao 1 2
Cao Jun 1 2
Ding Daqin 3
Hu Jinhui 1 2
Zhang Shuang 3
Kuang Li 1 2 kuangli0308@163.com
1 Department of Psychiatry, the First Affiliated Hospital of ChongQing Medical University , ChongQing, Yuzhong , China
2 Psychiatric Center, the First Affiliated Hospital of ChongQing Medical University , ChongQing, Yuzhong , China
3 Mental Health Center, University-Town Hospital of ChongQing Medical University , ChongQing, GaoXin , China
Di Tella Marialaura
Electronic publication date: 2025 Apr 4
Publication date: 2025
Volume: 13
Electronic Location ID: e19307
Received 2024 Nov 7; Accepted 2025 Mar 20
Copyright: © 2025 Ai et al.
Copyright year: 2025
Copyright holder: Ai et al.
License: This is an open access article distributed under the terms of the Creative Commons Attribution License, which permits unrestricted use, distribution, reproduction and adaptation in any medium and for any purpose provided that it is properly attributed. For attribution, the original author(s), title, publication source (PeerJ) and either DOI or URL of the article must be cited.
License URL: https://creativecommons.org/licenses/by/4.0/

Keywords: Parental marital status, Self-harm, Depression, Chinese primary school students, Classmate relationships

Funding: Joint Project of Science and Health in Chongqing Medical Research 2022MSXM058 Natural Science Foundation of Chongqing, China STB2022NSCQ-MSX0053 Youth Innovation in Future Medicine, Chongqing Medical University W0138 Chongqing Higher Education Teaching Reform Research Project 243397 The First Affiliated Hospital of Chongqing Medical University “Discipline Peak Plan” cyyy-xkdfjh-cgzh-202304 This work was supported by the Joint Project of Science and Health in Chongqing Medical Research (No. 2022MSXM058), the Natural Science Foundation of Chongqing, China (No. STB2022NSCQ-MSX0053), the Program for Youth Innovation in Future Medicine, Chongqing Medical University (No. W0138), the Chongqing Higher Education Teaching Reform Research Project (243397), The First Affiliated Hospital of Chongqing Medical University “Discipline Peak Plan” scientific and technological achievement transformation project (cyyy-xkdfjh-cgzh-202304). The funders had no role in study design, data collection and analysis, decision to publish, or preparation of the manuscript.

==============================
Background

Self-harm is an increasing global public health concern, with a growing prevalence in younger children. This study investigates the associations between parental marital status and self-harm behaviors among primary school students, with a focus on the mediating role of depressive symptoms and the moderating effect of classmate relationships.

Methods

A cross-sectional survey was conducted among 33,285 students (grades 3–6; mean age = 10.36 years) in the Shapingba District of Chongqing, China, from September to December 2020. Self-report measures included the Children’s Depression Inventory (CDI), general demographic data, self-harm behaviors, and parental marital status. Data were analyzed using SPSS 26.0 for descriptive statistics and Mplus 8.1 for structural equation modeling (SEM), assessing the effects of parental marital status on self-harm.

Results

The reporting rates of depression and self-harm in grades 3–6 of primary school are 16.3% and 12.7%, respectively. Parental separation exhibited a more pronounced overall impact on self-harm (β = 0.120) compared to divorce (β = 0.105). Positive classmate relationships mitigated the indirect effect of separation on self-harm mediated by depression, reducing it from 0.098 to 0.072. Additionally, these relationships attenuated the direct effect of divorce on self-harm, decreasing it from 0.088 to 0.043. Depression significantly mediates the relationship between parental separation and children’s self-harm, with direct and indirect effects accounting for 53% (β = 0.057) and 47% (β = 0.063) of the total effect, respectively.

Conclusion

The marital status of parents, especially in cases of separation, has a significant impact on self-harm behaviors among primary school students, with depression acting as a key mediating factor. Supportive classmate relationships can alleviate this effect, highlighting their importance in mental health interventions. These findings offer valuable insights for the development of policies aimed at reducing self-harm and enhancing psychological well-being among children.

Introduction

Self-harm involves deliberately injuring one’s body tissues, without any intention of suicide (Soomro & Kakhi, 2008; Knipe et al., 2022). This concept covers various behaviors that can inflict non-fatal physical damage, including cutting, burning, scratching, and hitting (Geulayov et al., 2019). Self-harm is a critical public health issue for children and adolescents around the world and is a strong indicator of future suicidal tendencies (Hawton, Saunders & O’Connor, 2012). Worryingly, self-harm behavior is becoming more common among younger age groups (DeVille et al., 2020; Cairns et al., 2019). A worldwide meta-analysis found that about 6.2% of preadolescents in community samples have engaged in self-harm at some point in their lives (Liu et al., 2022). A study involving 11,814 children aged 9 to 10 years in the United States found that the prevalence of Self-harm among this age group is 9.1% (DeVille et al., 2020). From 2006 to 2016, intentional poisonings among Australians aged 5 to 19 increased annually by 8.39%, indicating a rising trend in self-harm in this age group (Cairns et al., 2019). In Asia, a study conducted in India reported deliberate self-harm rates of 4.5% in adolescent boys and 3.2% in girls aged 10–19, while a cross-sectional survey in Japan found a 12-month prevalence of self-harm of 3.8% among students aged 12–18, with rates increasing with age (Endo et al., 2017; Sinha et al., 2021). Additionally, 29.3% (28.5–30.1) of Chinese primary school students are estimated to have engaged in self-harm at some point in their lives, with a yearly rising trend (Qu et al., 2023). In a 2-year longitudinal study of 2,744 junior and senior high school students from two rural schools in Sichuan Province, China, the prevalence of self-harm among adolescents was found to be 21.2%, while no related research has been conducted in Chongqing, which is also located in Southwest China (Lai et al., 2021). However, scholarly investigation in this domain, especially within the context of China, has been constrained by difficulties associated with the acquisition of extensive and comprehensive mental health survey data from primary school students. As a result, prior studies have often been constrained by small sample sizes, limited regional coverage, and inconsistencies in reported prevalence rates (Wang et al., 2021; Tang et al., 2018; Qu et al., 2023). The issue of self-harm among students has drawn considerable attention from both the Chinese government and all sectors of society (Wang et al., 2021; Tang et al., 2018; Qu et al., 2023).

The family plays a crucial role in a child’s development, and parental divorce or separation can have a significant detrimental effect on a child’s mental health. Individuals affected by parental divorce are at an increased risk for a range of mental health issues, including emotional and behavioral disorders, depression, anxiety, and poor academic performance. Children from divorced or single-parent households are particularly vulnerable, exhibiting higher rates of self-harm compared to those from intact families (Çaksen, 2022). Divorce is a highly stressful life event for the entire family, profoundly affecting children by undermining their sense of stability and security, which in turn leads to emotional challenges and difficulties in adapting to significant life changes (Spremo, 2020). Parental divorce significantly elevates the risk of mental health issues, such as depression, anxiety, and suicidal behavior, as well as substance-based addictions, particularly among children and adolescents (Auersperg et al., 2019). Divorce increases the risk of depression in adolescents, which is associated with higher rates of suicidal behaviors and self-harm compared to those living with both biological parents (Park & Park, 2020). Parental separation or divorce during childhood is significantly associated with increased anxiety, depression, suicidality, and self-harm among university students (Bhattarai et al., 2023). Parental separation is an independent predictor of self-harm in youths with bipolar disorders (Janiri et al., 2024). However, previous studies on the relationship between parental marital dynamics and emotional disorders, self-harm, and suicidal behaviors in children and adolescents have predominantly focused on parental separation or divorce as a generalized concept. There has been insufficient examination of the specific effects of varying degrees of negative marital relationships, such as separation (separated but not divorced) vs. divorce, on self-harm behaviors. Moreover, extant research has predominantly concentrated on the impact of parental divorce or separation on the development of emotional issues, such as depression and self-harm, among children and adolescents. Nevertheless, there is a paucity of comprehensive analyses examining the relationship between distinct parental marital statuses and how they contribute to the onset of depression and self-harm. It is particularly noteworthy that studies addressing this issue, especially in the context of primary school students, remain limited.

The school environment plays a pivotal role in students’ lives, providing a setting where they dedicate substantial time to academic pursuits, establish relationships with peers and educators, and cultivate vital social-emotional competencies. Friendships and peer relationships have a profound impact on young people’s development, well-being, and behavior, which can encompass issues such as self-harm ideation and behavior (Bilello et al., 2024). Constructive peer relationships can serve as a protective factor, mitigating the incidence of suicidal ideation among students (Lee et al., 2021). While friendships and peer relationships serve as important sources of support that may offer protection against self-harm, current evidence regarding their role remains nascent and inconclusive.

This research examines the associations between parental marital status, depression, and self-harm behaviors in the context of the increasing prevalence of self-harm among adolescents. The study seeks to explore the impact of various parental dynamics-specifically, separation vs. divorce-on self-harm outcomes and to evaluate the moderating influence of peer relationships. Through the analysis of these factors, the study aims to contribute theoretical support for the development of policies focused on the prevention and intervention of depression and self-harm in students, thereby providing valuable insights for governmental agencies, educational institutions, and families. Based on these considerations, the study hypothesizes that adverse parental marital status-particularly distinguishing between separation and divorce-negatively impacts self-harm behaviors in children. Depression is identified as a mediating factor in this relationship, while positive peer relationships are found to moderate the effect, thereby mitigating the risk of self-harm associated with unfavorable family dynamics (Fig. 1).

Figure 1 The proposed moderated-mediation model of the relationship between parental marital status and self-harm.

Materials and Methods

Study participants and data collection

This research utilized a cross-sectional survey to assess mental health among primary school students in the Shapingba District of Chongqing, following the methods detailed in Ai et al. (2025). Data were collected from September to December 2020, with interruptions during examinations and holidays. Students were categorized into high-risk and low-risk groups based on screening results, with follow-up interviews conducted by professional psychiatrists for those identified as high risk.

All primary school students in grades 3 to 6 in Shapingba District were targeted, while grades 1 and 2 were excluded due to their age. Of the 33,895 students approached, 33,285 (51.72% female, mean age = 10.36 years) completed the survey, resulting in a response rate of 98.8%. Informed consent was obtained from participants and guardians, and no financial incentives were provided. Sampling method in this investigation, the cluster sampling method was used, selecting all primary schools within a specific administrative district of Chongqing to participate in the survey.

Cluster sampling was employed, involving all primary schools in the district. Prior to the survey, training was provided to head teachers and school psychologists to facilitate the study. During data collection, trained personnel were present to assist participants and ensure proper flow. A technical support team was also available to address any issues during the online survey administration.

Data collection

Given the comprehension levels of primary school students, students in grades 1–2 were not included in the study, the demographic data collected included gender, age, grade, and family economic status.

In this study, self-harm behaviors were determined based on a positive answer to the question, “Have you ever hurt yourself when you were unhappy?” with the response options response “Yes” or “No”. The marital status of parents was assessed with the following item: “What is the marital status of your parents?” The response options included: (1) Normal marriage, (2) Separation (separated but not divorced), (3) Divorced. The relationships with classmates at school were evaluated with the question: “How are your relationships with your classmates at school?” The response options for this question were: (1) Good, (2) General, (3) Poor.

Additionally, depressive symptoms were measured using the Children’s Depression Inventory (CDI), along with information about relationships with classmates (categorized as poor, average, or good). In this study, the Cronbach’s alpha value for the CDI scale was 0.884, indicating good internal consistency (Twenge & Nolen-Hoeksema, 2002).

Analysis method

The data underwent preprocessing utilizing SPSS version 26.0, followed by structural equation modeling (SEM) conducted with Mplus version 8.1 for the analytical procedures. The objective of this analysis was to evaluate both the direct and indirect effects of parental marital status on self-harm behaviors. Additionally, the study investigated the mediating role of depression and the moderating influence of classmate relationships.

Statistical analysis

Descriptive statistics and correlations were first analyzed using SPSS version 26.0. Next, the mediating role of depression and the moderating influence of classmate relationships were examined through structural equation modeling (SEM) using Mplus version 8.1. To account for the influence of confounding variables, gender and age were controlled in all statistical analyses.

Procedure

Ethics approval for this study was granted by the Ethics Committee of the First Affiliated Hospital of Chongqing Medical University (approval number: 2020-879). Prior to the screening, written informed consent was obtained from the students’ parents. Additionally, students provided electronic consent at the outset of their participation in the survey. Participating students allocated approximately 30 min to complete the screening process. This involved logging into a designated website, where their information and the questionnaires had been pre-uploaded, and subsequently completing the screening online during their computer class. The results were exported to the background and encrypted. To encourage honest responses, participants were informed that their responses would be kept strictly confidential and that their participation was voluntary and that they could refuse to participate in the study at any time. This study conformed to the ethical guidelines of the 1975 Declaration of Helsinki.

Results

Descriptive statistics

The demographic data of all participants, illustrated in Table 1, highlights several concerning aspects regarding mental health and social interactions. In the demographic data, 12.7% of participants reported engaging in self-harm, while 16.3% experienced depressive symptoms. Regarding bullying, 38.7% reported being bullied and 6.5% experiencing bullying several times or more. Additionally, 39.7% of participants reported experiencing high study pressure. The mean age of participants was 10.36 years (SD = 1.24), with a balanced gender distribution comprising 48.2% males and 51.7% females. The relationships among classmates reveal that a significant majority of participants, 74.6%, reported having good relationships with their classmates. However, 23.6% indicated general relationships, and only a small percentage, 1.72%, reported having poor relationships. Concerning parental relationships, 85.6% of participants indicated normal relationships with their parents, whereas 4.3% experienced parental separation and 10.0% reported parental divorce. Additionally, 83.2% of participants have good relationships with their parents, indicating strong familial bonds. Meanwhile, 15.8% report average relationships, and only 0.9% experience poor relationships, suggesting negative parental ties are uncommon. Overall, most participants view their parental relationships positively.

Table 1 The demographic data of all participants.

Variables	N/Mean	%/Standard	
Age	10.36	1.24	
Sex			
Male	16,071	48.2	
Female	17,214	51.7	
Self-harm			
0	29,062	87.3	
1	4,223	12.7	
Depression			
0	27,854	83.6	
1	5,431	16.3	
Relationship with classmates			
Good classmate relationships	24,860	74.6	
General classmate relationship	7,853	23.6	
Poor classmate relationships	572	1.72	
Parental relationship			
Normal	28,523	85.6	
Separation	1,425	4.28	
Divorce	3,337	10.02	
Relationship with parents			
Good relationship with parents	27,698	83.20	
General relationship with parents	5,261	15.8	
Poor relationship with parents	326	0.9	
Bullied			
None	20,421	61.30	
Rarely	10,678	32.0	
Several times	1,417	4.2	
Often	769	2.3	
Study pressure			
Less study pressure	13,235	39.7	
Generally average study pressure	16,876	50.7	
High study pressure	3,174	9.5	

Table 2 presents the descriptive statistics and correlations among the study variables. Notably, self-harm is significantly positively correlated with depression (r = 0.38, p < 0.01), indicating that higher levels of depression are associated with greater instances of self-harm. There are also notable correlations between self-harm and parental relationships; self-harm correlates positively with parental separation (r = 0.11, p < 0.01) and divorce (r = 0.10, p < 0.01), suggesting a connection between these adverse familial situations and self-harming behaviors. When examining the relationships with classmates, a negative correlation is observed with self-harm (r = −0.16, p < 0.01), implying that poorer relationships with classmates are associated with higher instances of self-harm. Similar negative correlations are also present for depression and classmate relations (r = −0.28, p < 0.01), highlighting that more depressive symptoms correlate with worse social interactions among peers.

Table 2 Descriptive statistics and correlations of the study variables.

The frequency, proportion, and correlations for the measured variables.

	1	2	3	4	5	
1. Self-harm	–					
2. Depression	0.38**	–				
3. Separation	0.11**	0.17**	–			
4. Divorce	0.10**	0.12**	−0.07**	–		
5. Classmate relations	−0.16**	−0.28**	−0.10**	−0.08**	–	
Frequency	4,223	5,431	1,425	3,337	24,860	
Proportion (%)	12.7	16.3	4.3	10.0	74.7	
Note:

** p < 0.01.

Increased probability of self-harm in children from separated families

Figure 2 presents the path coefficients and standard errors for the model assessing the impact of marital status on self-harm. The path coefficients quantify the strength and direction of the relationships between the variables. Notably, separation exhibits a significant positive effect on self-harm, with a path coefficient of 0.197 and a standard error of 0.013, indicating a robust and statistically significant relationship (p < 0.01). Similarly, divorce demonstrates a significant positive effect on self-harm, with a path coefficient of 0.115 and a standard error of 0.008, further corroborating a statistically significant relationship (p < 0.01). To further characterize the magnitude of these effects, logistic regression analyses revealed substantial effect sizes: separation showed an odds ratio of 3.65 (95% CI [3.24–4.11]) with a standardized β of 0.78, while divorce demonstrated an odds ratio of 2.41 (95% CI [2.20–2.63]) with a standardized β of 0.79 (Table 3). These results imply that marital status is a strong predictor of self-harm, with separation exerting a particularly pronounced effect.

Figure 2 The impact of parental marital status on self-harm.

Table 3 Effects of separation and divorce on self-harm: logistic regression results.

Variable	β	OR	95% CI lower	95% CI upper	Standardized β	
Separation	1.29	3.65	3.24	4.11	0.78	
Divorce	0.87	2.41	2.20	2.63	0.79	

Testing for mediation effect of depression

To assess the mediating effect of depression on the relationship between parental marital status and self-harm, a series of analyses were conducted. The total effect of parental separation on self-harm is represented by a coefficient of 0.319 (p < 0.001), with a direct effect of 0.093 (p < 0.001) and an indirect effect through depression of 0.325 (p < 0.001), indicating that depression plays a significant mediating role in this relationship. For children from divorced families, the total effect is 0.167 (p < 0.001), with a direct effect of 0.062 (p < 0.001) and an indirect effect through depression of 0.167 (p < 0.001), which also indicates that depression is an important mediator in the connection between parental divorce and self-harm (Fig. 3).

Figure 3 The mediating effect of depression in the association between parent’s marriage and self-harm primary school students.

The results suggest that depression plays a significant mediating role in the relationship between parental marital status and self-harm behaviors, highlighting the need for targeted interventions that address familial dynamics.

The moderating effect of classmate relationships

The influence of classmate relationships on self-harm behaviors was examined as a moderating factor in the context of parental marital status. The results indicate that the effects of separation and divorce on self-harm, with depression as the mediating variable and classmate relations as the moderating variable. The results showed that separation had a significant positive effect on depression (path coefficient of 0.31), while divorce had a smaller effect on depression (path coefficient of 0.19). The effect of depression on self-harm was small (path coefficient of 0.09) and its statistical significance was weak. The effect of good classmates on self-harm is negative (path coefficient of −0.02), but this moderating effect is small and may not be significant (Fig. 4).

Figure 4 The mediating role of depression and the moderating effect of classmate relationships.

Finally, we further estimated the direct and indirect effects of the model. The indirect effect of parental separation on self-harm via depression was found to be lower with good classmate relations (indirect effect = 0.072, p < 0.001) compared to those without the moderator (indirect effect = 0.098, p < 0.001). Similarly, the indirect impact of depression on divorce was also lower with good classmate relations (indirect effect = 0.035, p < 0.001) than when there was no moderator (indirect effect = 0.059, p < 0.001). Moreover, the direct effect of parental marital status on self-harm varied with the level of classmate relations. The direct effect of divorce (direct effect = 0.043, p < 0.001) was lower with good classmate relations than without the moderator (direct effect = 0.088, p < 0.001) (Table 4).

Table 4 The moderation effects in direct and indirect effects of the model.

Variable	Estimate	SE	t-value	p-value	
IND2_LOW	0.098	0.005	21.117	<0.001	
IND2_HIG	0.072	0.004	17.284	<0.001	
IND3_LOW	0.059	0.004	16.78	<0.001	
IND3_HIG	0.035	0.003	14.142	<0.001	
DIR3_LOW	0.088	0.010	9.104	<0.001	
DIR3_HIG	0.043	0.007	6.173	<0.001	
TOT2_LOW	0.189	0.013	14.251	<0.001	
TOT2_HIGW	0.160	0.012	13.325	<0.001	
TOT3_LOW	0.148	0.01	14.399	<0.001	
TOT3_HIG	0.078	0.007	10.657	<0.001	
Note:

2-Separation; 3-Divorce; IND, Indirect Effect; DIR, Direct Effect; _LOW - Without Moderation Path; _HIG - With Moderation Path; TOT, Total Path.

These findings suggest that positive classmate relationships can mitigate the risk of self-harm associated with parental separation and divorce. The moderating role of peer interactions emphasizes the importance of fostering supportive friendships to enhance resilience in children facing familial challenges.

Discussion

There has been a notable rise in the incidence of self-harming behaviors, with a particularly concerning trend toward younger age groups. Of significant concern is the prevalence of self-injurious behaviors among primary school students, which are influenced by a combination of school, family, and individual factors. In this study, we identified a self-harm reporting rate of 12.7% among primary school students and examined the impact of parental marital status on self-harming behaviors. The results indicate that children from families experiencing parental separation due to marital discord are more likely to engage in self-harm compared to those from divorced families. Depressive symptoms emerged as a key mediating factor, contributing to the increased occurrence of self-harm. Moreover, the presence of positive classmate relationships was found to mitigate the adverse effects of negative marital status on self-harm through depressive symptoms. The study highlights the need for targeted interventions to cultivate supportive classmate relationships while taking into account family factors to reduce self-harm among primary school students.

In this study, the reported self-harm rate among primary school students was 12.7%. Although this rate is lower than those observed in certain regions, it nonetheless constitutes a significant concern. In contrast, self-harm rates in underdeveloped areas within the same southwestern region have been documented to reach as high as 47.0% (Yang et al., 2022; Liu et al., 2020). Furthermore, the estimated lifetime prevalence of self-harm among primary school students in China is 29.3% (Qu et al., 2023). When compared to our findings, self-harm rates among adolescents aged 10–19 across eight provinces are reported at 14.3% (Zhang et al., 2023; Chen et al., 2024). These statistics highlight that self-harm is a significant issue affecting various demographic groups, underscoring the necessity for targeted interventions.

In addition, the variations in the prevalence of self-harm may be attributed to a range of factors, including cultural background, socioeconomic conditions, and educational pressures (Steare et al., 2023; Tang et al., 2018). Regions characterized by higher socioeconomic status may impose greater academic pressure on students, potentially leading to increased rates of self-harm behaviors (Chen et al., 2024; Wang et al., 2021). Furthermore, family support systems are crucial in influencing self-harm behaviors, particularly among left-behind children. In Yunnan Province, the prevalence of self-harm among left-behind children has been reported to reach as high as 48.8% (Tian et al., 2019), highlighting the heightened risk faced by this demographic and emphasizing the necessity for targeted intervention strategies.

Furthermore, this study indicates that the probability of self-harm among primary school students is higher in cases of parental separation than in instances of divorce. Both separation and divorce, arising from marital discord, are generally regarded as adverse marital conditions. A considerable body of research has demonstrated a significant correlation between such negative marital statuses and elevated rates of depression, self-harm, and suicidal behaviors among children (Storksen et al., 2007; Afifi et al., 2009; Spremo, 2020). Existing research predominantly examines the influence of parental marital status, whether beneficial or detrimental, on children’s mental health outcomes, including suicidal ideation and behaviors (Alonzo et al., 2014; Sands, Thompson & Gaysina, 2017). There is, however, a paucity of studies investigating the differential impacts of various negative marital relationship forms, such as separation vs. divorce, on children’s self-harm behaviors. Furthermore, the majority of these studies focus on adolescents aged 10 to 25, resulting in a significant research gap concerning children under the age of 12 (McEvoy et al., 2023; Victor et al., 2019). Our study corroborates the adverse effects of negative marital statuses on children’s mental health and further elucidates the differential impacts of distinct types of negative marital relationships on self-harm behaviors in children. The analysis demonstrates that the prevalence of self-harm (0.11:0.10) and depression (0.17:0.12) is greater among primary school students from separated families compared to those from divorced families. This finding challenges Hypothesis 1, indicating that self-harm among primary school students is not exclusively determined by a simplistic evaluation of positive or negative parental relationships. Rather, it highlights the necessity for increased focus on the mental health of children from families undergoing separation due to parental discord.

Notably, younger children’s limited understanding of marital transitions introduces complexity in temporal analysis, with 10% of elementary students in our study unaware of ongoing parental separations. This phenomenon aligns with “protective concealment” strategies—where parents intentionally withhold divorce information or maintain de facto separations to shield children from distress (Du et al., 2022; Tafà et al., 2022). Importantly, childhood self-harm may itself serve as an indicator of pre-existing familial dysfunction, consistent with the spill-over hypothesis (Stover et al., 2016), where unresolved marital conflicts disrupt parent-child interactions and trigger maladaptive behaviors (O’Hara et al., 2019). These findings underscore the need for longitudinal multi-informant studies to clarify the bidirectional dynamics between marital discord and self-harm.

Furthermore, this study found that the mediating role of depression is particularly pronounced among primary school students. A significant correlation exists between parental divorce and the prevalence of depression, suicide attempts, and suicidal ideation in children (Auersperg et al., 2019; Sands, Thompson & Gaysina, 2017; Jiang et al., 2024; Watanabe et al., 2012; Victor et al., 2019). This study identified a reported depression rate of 16.3% among primary school students. Furthermore, the indirect effects of depression on self-harm in students from separated and divorced families were quantified as 0.063 and 0.049, respectively. This indicates that an atypical parental marital status is correlated with an increased prevalence of depressive symptoms among primary school students, thereby elevating the risk of self-harming behaviors (Spremo, 2020). These students frequently lack the ability to foresee the transition from separation to divorce, as well as the subsequent ramifications, which may include estrangement from close family members, alterations in educational environments, relocation, and modifications in lifestyle (Çaksen, 2022). The mediating role of depression is notably significant. Previous research has demonstrated that depression exacerbates the incidence of self-harm; however, the current study reveals that the mediating effect of depression accounts for approximately 50% of this relationship (Salt, Crofford & Segerstrom, 2017). This finding highlights the critical importance of considering depression as a substantial mediating variable influencing the occurrence of self-harm among primary school students.

Lastly, the moderating influence of positive classmate relationships is clearly demonstrated in the context of parental marital status, depressive symptoms, and self-harm behaviors, where they play a multifaceted role. This role can be either protective or risk-enhancing. Prior research has indicated that inadequate classmate relationships may intensify self-harm behaviors (Auersperg et al., 2019). The findings from the moderation model reveal that positive classmate relationships are significantly and inversely associated with both depression (effect size = −0.234) and self-harm behavior (effect size = −0.048).The findings suggest that strong classmate relationships can substantially mitigate the incidence of depressive symptoms and self-harming behaviors. This indicates that good classmate relationships can significantly reduce the occurrence of depressive symptoms and self-harm behavior. Furthermore, positive classmate relationships can decrease the risk of depression caused by parental marital discord by more than 20%, and reduce the probability of depression triggered by parental divorce by more than 30%, thereby further reducing the occurrence of self-harm behavior. This result emphasizes the key role of classmate relationships in buffering the negative impact of adverse family environments on mental health (Steinhoff et al., 2023).

Limitations and future directions

This study is constrained by several limitations. Primarily, it employs a cross-sectional research design, which restricts the capacity to establish causal relationships. Future research should consider adopting longitudinal designs to facilitate a more comprehensive understanding of these causal connections, as well as distinguish the dynamic relationship between “marital problems lead to self-harm” and “self-harm aggravates marriage breakdown”. Second, the sample is restricted to urban primary school students in China, which may limit the generalizability of the findings. Future studies are encouraged to conduct multi-center research to improve the diversity and broader applicability of the results. Third, the reliance on self-report questionnaires may introduce recall bias. Future research should consider incorporating a variety of assessment methods, such as peer-reported questionnaires and semi-structured interviews, to enhance the reliability of the data.

Conclusions

This study primarily examines self-harm behaviors among primary school students under the age of 12, a demographic that has been largely overlooked in previous research predominantly focused on adolescents. It provides significant insights into self-harm behaviors in this group, particularly highlighting the elevated risk faced by children from separated families in contrast to those from divorced families. The findings of our study suggest that depression acts as a principal mediating factor affecting self-harm, whereas positive relationships with classmates function as a significant protective factor. These outcomes emphasize the critical importance of addressing mental health in children, particularly within the context of parental marital challenges. Furthermore, they highlight the need for targeted interventions designed to cultivate supportive relationships among classmates to improve overall well-being, thereby corroborating our research hypothesis. Future research should prioritize the development of effective strategies to mitigate the psychological impacts of parental separation and divorce. By gaining a deeper understanding of these mechanisms, we can improve our capacity to support children experiencing parental marital transitions and reduce the incidence of depression and self-harming behaviors.

Supplemental Information

Supplemental Information 1 Original data on parental marital status and self-harm in primary school students.

Supplemental Information 2 Mental Health questionnaire of primary school student (Chinese).

Supplemental Information 3 Mental Health questionnaire of primary school student (English).

Supplemental Information 4 Codebook.

We sincerely thank the Shapingba District Education Committee in Chongqing for their ongoing support. We are also grateful to the students, doctors, and participants for their invaluable contributions, and to Chengdu Knowledge Vision Technology Co., Ltd. for their essential help with data management. Special thanks to the Xin Meiao Health Management Institute for their significant role in data collection and organization.

Additional Information and Declarations

Competing Interests

The authors declare that they have no competing interests.

Author Contributions

Ming Ai conceived and designed the experiments, analyzed the data, prepared figures and/or tables, authored or reviewed drafts of the article, and approved the final draft.

Xiao-Ming Xu conceived and designed the experiments, performed the experiments, analyzed the data, prepared figures and/or tables, authored or reviewed drafts of the article, and approved the final draft.

Wo Wang conceived and designed the experiments, performed the experiments, prepared figures and/or tables, authored or reviewed drafts of the article, and approved the final draft.

JianMei Chen conceived and designed the experiments, analyzed the data, authored or reviewed drafts of the article, and approved the final draft.

Jinglan He performed the experiments, analyzed the data, prepared figures and/or tables, and approved the final draft.

Qi Zhang performed the experiments, analyzed the data, prepared figures and/or tables, and approved the final draft.

Su Hong conceived and designed the experiments, performed the experiments, analyzed the data, prepared figures and/or tables, authored or reviewed drafts of the article, and approved the final draft.

Yao Gan performed the experiments, analyzed the data, prepared figures and/or tables, authored or reviewed drafts of the article, and approved the final draft.

Jun Cao analyzed the data, prepared figures and/or tables, and approved the final draft.

Daqin Ding analyzed the data, prepared figures and/or tables, and approved the final draft.

Jinhui Hu analyzed the data, authored or reviewed drafts of the article, and approved the final draft.

Shuang Zhang performed the experiments, analyzed the data, prepared figures and/or tables, authored or reviewed drafts of the article, and approved the final draft.

Li Kuang conceived and designed the experiments, authored or reviewed drafts of the article, and approved the final draft.

Human Ethics

The following information was supplied relating to ethical approvals (i.e., approving body and any reference numbers):

Ethics approval for this study was granted by the Ethics Committee of the First Affiliated Hospital of Chongqing Medical University (approval number: 2020-879).

Data Availability

The following information was supplied regarding data availability:

The raw data is available in the Supplemental File.

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
