# Peer review of "Impact of parental marital status on self-harm in Chinese primary school students: the mediating role of depression and the moderating effect of classmate relationships"

_PeerJ, doi:10.7717/peerj.19307_

## Round 0.1 · original submission · Major Revisions

Dear Authors,

Thank you for submitting your manuscript to PeerJ.
The Reviewers have highlighted some issues that need to be addressed. Please revise your paper taking into consideration all the Reviewers' comments and suggestions.

Kind regards,
Marialaura Di Tella

·

Basic reporting

The scientific background for investigating the impact of separation or divorce separately on self-harm behaviors, the mediating effect of depression, and the moderating effect of peer relationships is unclear. The authors stated that these impacts or effects have not sufficiently been examined. The details of what previous studies have shown and what are the limitations of previous studies may need to be indicated. In addition, the scientific reasons for investigating these factors may need to be stated.

The authors propose three hypotheses. However, the process of generating the three hypotheses is unclear.

Experimental design

Self-harm behaviors were assessed by the question: “Have you ever hurt yourself when you were unhappy?” Thus, the timing of self-harm behaviors may be preceded by the timing of separation or divorce, and it may not be possible to examine the effect of parental marital status on self-harm behaviors using this study design.

Validity of the findings

The details of what previous studies have shown and the limitations of previous studies are not shown. Thus, the Impact of this study is unclear.

Additional comments

Methods
All dependent variables in the SEM models were dichotomous. Thus, a logistic regression may be appropriate for these models.

Results
“This suggests that both parental marital statuses are strong predictors of self-harm, with separation being particularly pronounced impact. (Figure 1).”
The results may not indicate the strength of the effect of marital status on self-harm.

“These findings indicate that depression significantly mediates the impact of parental marital status on self-harm behaviors, underscoring the necessity for targeted interventions that address both familial dynamics and mental health in children.”
Depression partially mediates the association between marital status and self-harm behaviors. Thus, targeted interventions that address familial dynamics may be needed, but those that address mental health may not be necessary.

“For children from separated families, the direct effect on self-harm is 0.060 (p < 0.001) when peer relationships are low and 0.048 (p < 0.001) when they are high. Similarly, for children from divorced families, the direct effect is 0.053 (p < 0.001) for low peer relations and 0.036 (p < 0.001) for high peer relations (Figure 2).”
How were these effects obtained from the results shown in Figure 2?

Table 1
What does the frequency of classmate relations indicate?

The correlation between divorced and separated family was −0.07. This should be 0.

Reviewer 2 ·

Basic reporting

Start the introduction by giving a brief background on the problem being studied; (Problems in other countries, then in China, to where the research is being conducted). This will make it clearer why the research is important.

The authors should provide a more detailed explanation of the theoretical framework underpinning this research. This is essential to provide a clear context for the reader and to demonstrate how this study contributes to the advancement of knowledge in this field

Experimental design

Please discuss how the researchers ensured that the sample was representative of the target population. What criteria were used to select participants. What sampling method was used?

Validity of the findings

No Comment

Additional comments

Detailed comments on the manuscript can be found in the attached annotated version of the article

Annotated reviews are not available for download in order to protect the identity of reviewers who chose to remain anonymous.

Reviewer 3 ·

Basic reporting

1. Please specify "Chinease primary school students" in the title.
2. Please recheck the abstract pattern. Is any keyword needed?
3. In the introduction, please identify the results you can imply for clinical practice and education policy.
4. Regarding the hypotheses, please provide a raw figure to make it more understandable.

Experimental design

1. Line 103-107, please describe the demographic data of the results part.
2. Please describe the method you conducted for statistical analysis.

Validity of the findings

1. Please give more demographic data of the participants
2. The results are very short; give more details for each objective.
3. The discussion is too lengthy. Please compare your results with the previous studies, not to review all previous studies. (Line 201-220) the author should focus on clinical implications.
4. The highlighted part repeats the discussion; please enhance the strength of your study and its potential impact on policy.
5. The conclusion parts, please reply to all objectives, primary and secondary objectives. The reference is not necessary.

Additional comments

1. Please re-update the reference to 2014-2024, except for the classic models and measurement validation.
2. Please create the table 1, for the demographic data and correlation with self-harm and depression (p-value shown)

---

## Round 0.2 · Minor Revisions

Dear Authors,

Thank you for submitting your manuscript to Peer J.
The Reviewers have highlighted some additional aspects that should be revised before the paper can be accepted for publication. Please follow the Reviewers' comments and suggestions and revise your manuscript accordingly.

Kind regards,
Marialaura Di Tella

·

Basic reporting

“Response 2:
We appreciate the reviewer’s insightful feedback regarding the assessment of self-harm behaviors. We acknowledge the limitation that the timing of self-harm behaviors may indeed precede the events of separation or divorce. However, these instances that occurred before the parents' divorce or separation represent only a small portion of the data. Given the large sample size of our study, we believe that the overall findings regarding the relationship between parental marital status and self-harm behaviors remain valid and robust.”

Please add the reasons for expecting that a small portion of the students had self-harm behaviors before the events of separation or divorce in the Discussion.


“Response 5:”
“These results imply that marital status is a strong predictor of self-harm, with separation exerting a particularly pronounced effect.”
It remains unclear why the results imply that marital status is a “strong” predictor. Was the effect size calculated?

Table 3
Please add details of the variables shown in this table.

Experimental design

no comment

Validity of the findings

no comment

Additional comments

no comment

Reviewer 3 ·

Basic reporting

The revision is good.

Experimental design

No revision is needed.

Validity of the findings

The revision is good.

Additional comments

None

---

## Round 0.3 · accepted · Accept

Dear Authors,

Thank you for submitting your manuscript to Peer J.
The Reviewers' comments and suggestions have been adequately addressed. Therefore, the paper can be accepted for publication.

Kind regards,
Marialaura Di Tella